# A Long-Term Cloud Albedo Data Record Since 1980 from UV Satellite Sensors

**Clark J. Weaver [1],\*, Dong L. Wu [2], Pawan K. Bhartia [3], Gordon J. Labow [4] and David P. Haffner [4]**

[1] Earth System Science Interdisciplinary Center (ESSIC), University of Maryland,
College Park, MD 20742, USA
[2] Climate and Radiation Laboratory, NASA Goddard Space Flight Center, Greenbelt, MD 20771, USA;
dong.l.wu@nasa.gov
[3] Atmospheric Chemistry and Dynamics Branch, NASA Goddard Space Flight Center,
Greenbelt, MD 20771, USA; pawan.k.bhartia@nasa.gov
[4] Science Systems and Applications (SSAI), Inc., Lanham, MD 20706, USA; gordon.labow@ssaihq.com (G.L.);
david.haffner@ssaihq.com (D.P.H.)
**\*** Correspondence: clark.j.weaver@nasa.gov

**Abstract:** Black-sky cloud albedo (BCA) is derived from satellite UV 340 nm observations from NOAA and NASA satellites to infer long-term (1980–2018) shortwave cloud albedo variations induced by volcano eruptions, the El Niño–Southern Oscillation, and decadal warming. While the UV cloud albedo has shown no long-term trend since 1980, there are statistically significant reductions over the North Atlantic and over the marine stratocumulus decks off the coast of California; increases in cloud albedo can be seen over Southeast Asia and over cloud decks off the coast of South America. The derived BCA assumes a C-1 water cloud model with varying cloud optical depths and a Cox–Munk surface BRDF over the ocean, using radiances calibrated over the East Antarctic Plateau and Greenland ice sheets during summer.

**Keywords:** SBUV; OMPS; cloud albedo; diurnal cycle; cloud feedback; ENSO; volcanoes

## 1. Introduction

Uncertainty in the shortwave (SW) cloud radiative feedback to surface temperature variations is a key uncertainty in global temperature prediction for the future climate [1,2]. Climate models disagree on the magnitude, distribution, and processes of SW cloud albedo responses to a warming climate. A long-term, well-calibrated observational record of cloud albedo can help reduce this inter-model disparity by enabling the evaluation of historical and Atmospheric Model Intercomparison Project (AMIP) simulations from the Climate Model Inter-Comparison Project (CMIP).

Long-term, reliably calibrated satellite flux data are still lacking because of challenges to inter-calibrate observed radiances from instruments on different satellites. The longest calibrated flux record is from the Clouds and the Earth's Radiant Energy System (CERES; 2000 to present [3,4]) of which the SW flux covers 0.3–5 µm. Both the CERES shortwave and total radiation channels have onboard calibration sources to track the sensor stability and degradation. While the 20-year span of the CERES data record includes two major El Niño–Southern Oscillation (ENSO) events, it does not include the two most recent major volcanic eruptions.

A longer 39-year record of UV measurements from NASA and NOAA satellites does allow the study of these very important pre-CERES events. Previous work by Herman et al. [5] and Weaver et al. [6] construct a record of the Lambertian equivalent reflectivity (LER) from the UV measurements. LER is the reflectivity derived for the Earth's surface, bounding a purely Rayleigh atmosphere, consistent with

measured the top of atmosphere (TOA) radiance; an assumption is that the surface is Lambertian, and the effects of aerosols and clouds are included in the LER of the scene. Although it is only a proxy for the SW flux at the top of atmosphere (TOA), the longer-term UV record includes the El Chichón and Mt. Pinatubo volcanic events and three additional ENSO events, and covers two decades (1980–2000), when global temperatures warmed 0.3 °C [7,8].

The measured intensities used in this study, as well as in the previous LER studies, are observed by the Solar Backscatter UV (SBUV) instruments onboard the Nimbus-7, NOAA-9, -11, -14, -16, -17, -18, and -19 spacecraft [9,10]. After 2012, the record includes measurements from the nadir mapper (NM) instrument of the Suomi NPP Ozone Mapping Profiler Suite (OMPS) [11]. Weaver et al. [12] detail the inter-calibration of these SBUV instruments over the Antarctic and Greenland ice sheets and show that the radiances can be radiometrically calibrated to within a 2-σ uncertainty of 0.35%.

In this study, we improve the earlier analysis of LER by now accounting for non-Lambertian effects of the underlying scene. The black-sky cloud albedo (BCA) from measured UV radiances should more accurately account for the BRDF of the clouds and the underlying ocean/land surface than LER. LER estimates over cloud-free scenes near the glint angle can be highly biased because of the additional surface contribution. Such a BCA calculation is able to isolate the cloud signal and exclude ocean surface reflection. An additional refinement is an empirical adjustment that accounts for the cloud diurnal cycle.

## 2. Black-Sky Cloud Albedo

Black-sky cloud albedo (BCA) is defined as albedo in the absence of a diffuse component and is a function of illumination angle. For our retrieval, BCA was the ratio of hemispherically averaged upwelling TOA flux from clouds and aerosols to the downwelling flux scaled by 100. We removed the diffuse Rayleigh component and used the solar zenith angle at noon for the latitude and day of the measurement. The contribution from the underlying surface and Rayleigh scattering were estimated by the vector linearized discrete ordinate radiative transfer package (VLIDORT) model (Spurr, 2006 [13]) and were removed. The BCA was calculated from narrowband (~1 nm) backscattered intensities at 340 nm measured by the UV sensing satellite instruments. Regular sun-viewing irradiance measurements ($F_{sun}$) were made, typically weekly, to provide long-term calibration information. The measured intensities were normalized by $F_{sun}$, and multiplied by π. Throughout this study, I refers to the sun's normalized intensities. We corrected the intensities for the small ozone absorption, based on the reported total column ozone [14].

I varied between 0 and 1 if the reflecting surface is Lambertian and there is no absorption. We produced a 39-year record of BCA from these normalized intensities. Specifically, BCA is ratio of hemispherically averaged upwelling TOA flux from clouds and aerosols, $F^{up}_{cloud}$ to the downwelling flux at the TOA, $F^{down}$ multiplied by 100. The calculation of BCA is at the satellite pixel level, using the normalized intensity for a 150 × 150 km SBUV field of view (FOV). For OMPS, the intensities of the smaller 50 × 50 km FOV are used.

The VLIDORT accounts for contributions from clouds, aerosols, Rayleigh molecular scattering, surface reflectivity, and absorption from ocean chlorophyll, which all contribute to the observed normalized intensities ($I_{obs}$). We simulated $I_{obs}$ and finally produced the cloud signal in terms of BCA. The VLIDORT package offers several options to account for ocean sun glint. All are based on probability distributions of slopes of wind-driven ocean waves obtained by Cox and Munk [15,16] from photographs of sun glitter. The New-GISS Cox–Munk appeared to best simulate the $I_{obs}$ at low glint angles in the UV. To account for surface wind speed, we use the Goddard Modeling and Assimilation Office MERRA-2 product [17] from 1980 to present. VLIDORT also simulates the water leaving radiance determined by chlorophyll concentrations; here we used geographically varying monthly climatological values.

Over land, we assumed a constant Lambertian surface reflectivity (4%), which does not account for the known geographic variability: 2% for some grasslands and forest regions to 14% for some sandy desert regions [18].

We used a C-1 water cloud droplet size distribution [19] with an effective radius of 10 mm and a cloud top/bottom of 750/800 hPa, respectively. In the strictest sense, this model is valid only for stratocumulus water clouds of moderate optical thickness. Calculated intensities are only ~3% lower for an effective radius up to 30 mm, and there is negligible sensitivity to cloud-top height. VLIDORT calculations show differences in intensities over C-1 clouds and ice clouds with the same cloud optical depth (COD), but we had no independent information to distinguish the phases, let alone crystal shapes.

### 2.1. Determination of COD from $I_{obs}$

The first part of the BCA calculation determined the cloud optical depth (COD) consistent with the observed normalized intensities. VLIDORT calculated the normalized intensity that the instrument would measure based on the satellite viewing geometry for varying levels of COD. Using VLIDORT pre-calculated look-up tables, we retrieved the COD for the instrument FOV, assuming that the FOV was fully homogeneous.

### 2.2. Determination of Hemispherically-Integrated Flux from Cloud Optical Depth

Once a scene's COD was determined, we used another set of pre-calculated look-up tables to determine the TOA hemispherically-integrated upward flux ($F^{up}$) for varying levels of COD. We were only interested in the cloud signal, so the Rayleigh scattering was turned off for the calculation of these VLIDORT look-up tables. The $F^{up}$, which equaled the flux from the cloud, $F^{up}_{cloud}$, plus the underlying surface $F^{up}_{s}$, was determined from the COD. To isolate the cloud signal, the surface contribution was removed (Equation (1)):

$$F^{up}_{cloud} = F^{up} - F^{up}_{s} \tag{1}$$

A given underlying surface and cloud configuration can have many $F^{up}$ values that depend on the illumination angle. We used the solar zenith angle at local noon for the latitude and day of the measurement for the $F^{up}$ calculations. Note that in Section 2.1, we use the solar zenith angle at the actual time (rather than at noon) of measurement.

The same approach was used over land, except that a Lambertian surface with 4% albedo was assumed everywhere. Over land, BCA was entirely dependent on $I_{obs}$. Over ocean, the time-dependent MERRA2 winds and a time invariant climatology of chlorophyll concentrations were used to determine the level of cloudiness (COD), but this surface contribution was removed (Equation (1)).

## 3. Zonal Mean SBUV Time Series

For normal operations, a single SBUV instrument is able to fully sample a 15°-wide latitude band during a single month; sampling locations are evenly distributed over each latitude band. Since there are instances of reduced sampling, we checked that 50% of each 2.5° × 2.5° grid cell in the latitude band had at least 2 samples, and rejected those with insufficient coverage. In addition, BCA values observed when the solar zenith angle was above 75° were excluded from the zonal averages.

### 3.1. Inter-Satellite Differences

There are often significant inter-satellite differences when two or more SBUV instruments are sampling a latitude band. While all bands showed discrepancies between individual SBUV instruments, 30°S to 15°S had the most disagreement, showing differences up to 5% BCA (Figure 1a). We first noticed these inter-satellite differences when analyzing LER, and thought that improving the scene BRDF with the more realistic assumptions inherent with BCA would reduce the differences. But this was not the case; a time series of LER also showed the same differences and looked very similar to Figure 1a.

Possible culprits were the simplified assumptions used to derive BCA, or not accounting for important physical mechanisms. For example, our BCA retrieval uses radiative transfer calculations that assume a fixed C-1 water droplet distribution globally. Ice clouds or water clouds with a different effective radius than C-1 will retrieve inaccurate BCA. Secondly, our plane parallel (1-D) radiative transfer model could not account for 3D effects in the real atmosphere. Finally, changes in BCA due to the cloud diurnal cycle could also explain the inter-satellite differences. These issues are addressed below.

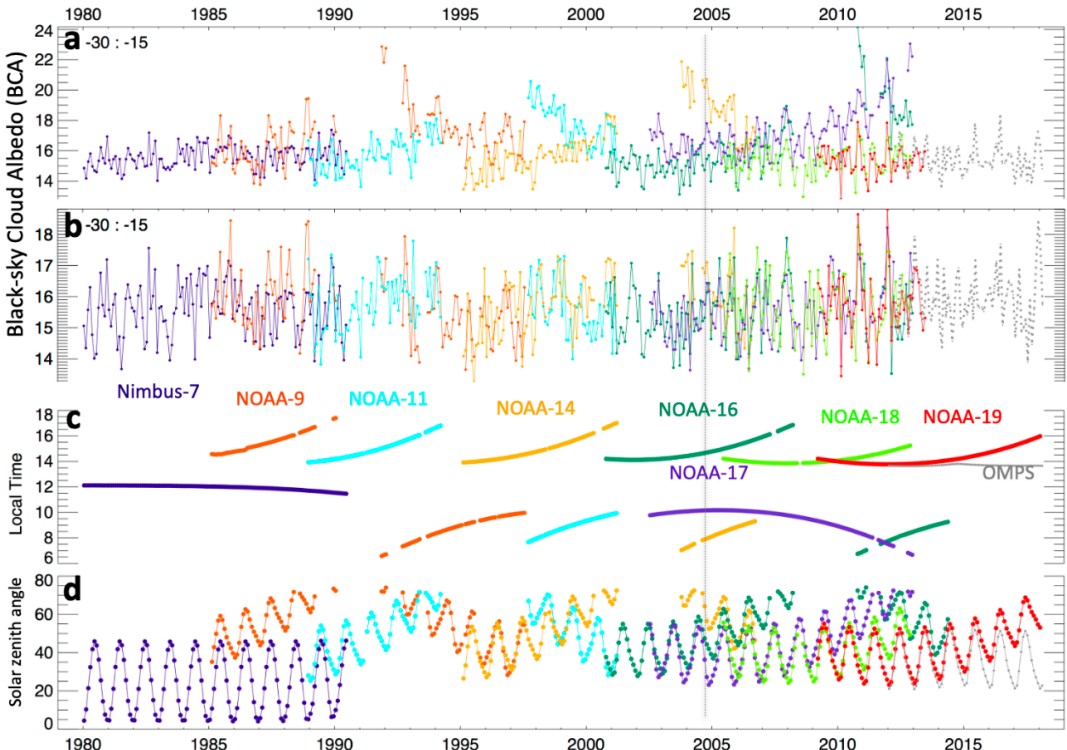

**Figure 1.** Monthly mean values of black-sky cloud albedo (BCA) (%) over ocean from individual solar backscatter UV (SBUV) instruments (colored traces) and the Ozone Mapping Profiler Suite (OMPS) mapper (gray dotted traces) for the 30°S to 15°S latitude band. (**a**) Un-adjusted BCA, (**b**) diurnally adjusted BCA, (**c**) local time, and (**d**) solar zenith angle.

A number of studies have shown that cloud optical depths retrieved from satellite observations increase with solar zenith angle, especially at higher angles [20–23]. We saw similar behavior; the most egregiously high BCA values were often associated with high solar zenith angles (Figure 1d). This has been attributed to radiative transfer differences between a plane parallel (1D) atmosphere, assumed in cloud retrievals, and the 3D atmosphere in the real world [24,25]. This mechanism also contributes to the inter-satellite differences, and is discussed later with the diurnal adjustment in Section 3.2.

The drifting orbits of the NOAA SBUV satellite platforms mean that any latitude band can be sampled at multiple local times during daylight hours (Figure 1c). If a location has a significant cloud diurnal cycle and sampling is at different times of the day, there will be BCA differences. In late 2004, the BCA estimates from NOAA-14, -17, and -16 were sampled at 8:00 a.m., 10 a.m., and 2:30 p.m., with the highest BCA in the early morning and the lowest values in the afternoon. This is consistent with the diurnal cycle for stratocumulus clouds, which are prevalent between 30°S to 15°S, [26,27]. Previous analysis of LER by Labow et al. [28] showed a diurnal variation attributed to the cloud daily life cycle; further, they describe a statistical approach to adjust an LER measurement made any time during the day to a noontime value. Specifically, Labow et al. gather the LER statistics for each 5° latitude band and one-hour local time bin and then construct a table of mean LER values for the latitude bands and local times. This is done for ocean and land locations separately. The noon-time adjustment to

any LER measurement at the FOV level is simply the difference between the table LER at the time of measurement minus the table LER at noon.

### 3.2. Diurnal Adjustment Based on Frequency Distributions

Our diurnal correction built on this approach, with adjustments to BCA based on frequency distributions rather than single value averages. Specifically, we derived a diurnal adjustment ($\Delta BCA_{diurnal}$) that, when subtracted from each BCA FOV observation ($BCA_{obs}$), was an estimate of the BCA that would have been measured at a reference local time earlier or later on that day ($BCA_{ref}$):

$$BCA_{ref} = BCA_{obs} - \Delta BCA_{diurnal} \tag{2}$$

The diurnal adjustments were derived separately for geographic locations with maximum cloudiness in the morning (morning-peaking) and with maximum in the afternoon. To determine these locations, we generated a climatology of morning minus afternoon BCA from the suite of eight SBUV instruments (Figure 2a). Each SBUV observation at $150 \times 150$ km FOV had a retrieved BCA, a local time, solar zenith angle, and geographic location. This information was binned by 9 local time bins from 5:15 a.m. to 6:45 p.m. and gridded by 1° latitude $\times$ 2° longitude. For each grid cell, we subtracted the mean BCA of morning bins (8:15–11:15 a.m.) from the afternoon (12:45–3:45 p.m.) bins. The resulting map clearly separates geographic locations where it is cloudier in the morning (red grid cells) from locations with clouds peaking in the afternoon (blue cells). Over ocean, areas with more morning clouds occur where low-level marine stratocumulus frequently exist (red grid cells) and are distinct from the afternoon peaking clouds that occur where higher altitude marine clouds prevail (blue cells). Over land, BCA typically peaks in the afternoon. Figure 2a is very similar to a map of the difference in cloud fraction between 10:30 a.m. and 1:30 p.m. local time as measured by Terra and Aqua MODIS instruments shown in Figure 2 of Labow et al. [28].

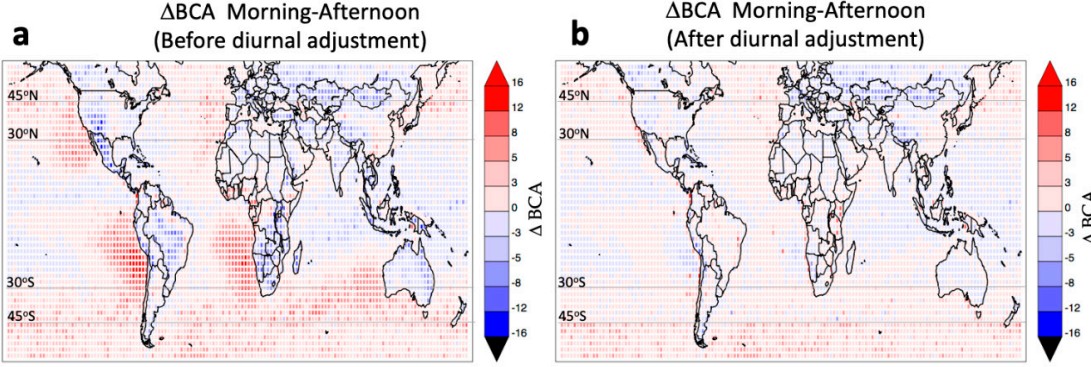

**Figure 2.** BCA difference between morning (8:15 a.m. to 11:15 a.m. local time) and afternoon (12:45 p.m. to 3:45 p.m.) for (**a**) un-adjusted BCA, and (**b**) after diurnal adjustment.

By collecting separate statistics for the morning and afternoon peaking clouds, we derived separate frequency distributions for the low-level marine stratocumulus and the higher altitude clouds. Since mid-latitude and tropical clouds might have had different diurnal cycles, we also separated by 5 latitude bands (45°S to 30°S, 30°S to 15°S, 15°S to 15°N, 15°N to 30°N, and 30°N to 45°N) and separated ocean from land locations.

Our diurnal statistics were based on 14 million observations sampled by the suite of eight SBUV instruments since 1980. First, we sorted by ocean and land. Second, based on an observation's latitude and longitude, we determined if the clouds in the FOV were morning- or afternoon-peaking, based on the climatology shown in Figure 2a. Third, we sorted by latitude using the 5 latitude bands. If the observation was off the coast of Peru, then it would be classified in the "ocean, morning-peaking, 30°S to 15°S" geographic bin; an observation over Colorado would be "land, afternoon-peaking,

30°N to 45°N". At this stage, this sorting was based only on geographic location; therefore, we had 2 (ocean–land) × 2 (morning- afternoon-peaking clouds) × 5 (latitude bands), which equaled 20 geographic bins. Within each geographic bin, we further sorted by the local time of the observation using nine local time bins and finally sorted by the solar zenith angle of the observation using eight solar zenith angle bins. To summarize, we had 20 geographic bins; within each geographic bin there were 9 (local time) × 8 (solar zenith angle) bins for a total of 1440 bins.

Histograms (normalized frequency distribution) of BCA for the ocean, morning-peaking 45°S to 30°S and 30°S to 15°S geographic bins both show the morning cloud diurnal cycle (Figure 3). The most likely time for high values of BCA were in the early morning and the most likely time for low values of BCA were late afternoon. Note that within any local time bin, the BCA frequencies generally shifted to higher values with increasing solar zenith angle; this was most apparent for the 45°S to 30°S band (Figure 3a). The intent of the diurnal adjustment was to modify the distributions in each bin to look like a reference distribution. It was constructed by averaging the 9:45–11:15 a.m. and the 12:45–2:45 p.m. bins, and was representative of a noon local time frequency distribution. The reference solar zenith angle was the local noon time solar zenith angle at the latitude of the observation.

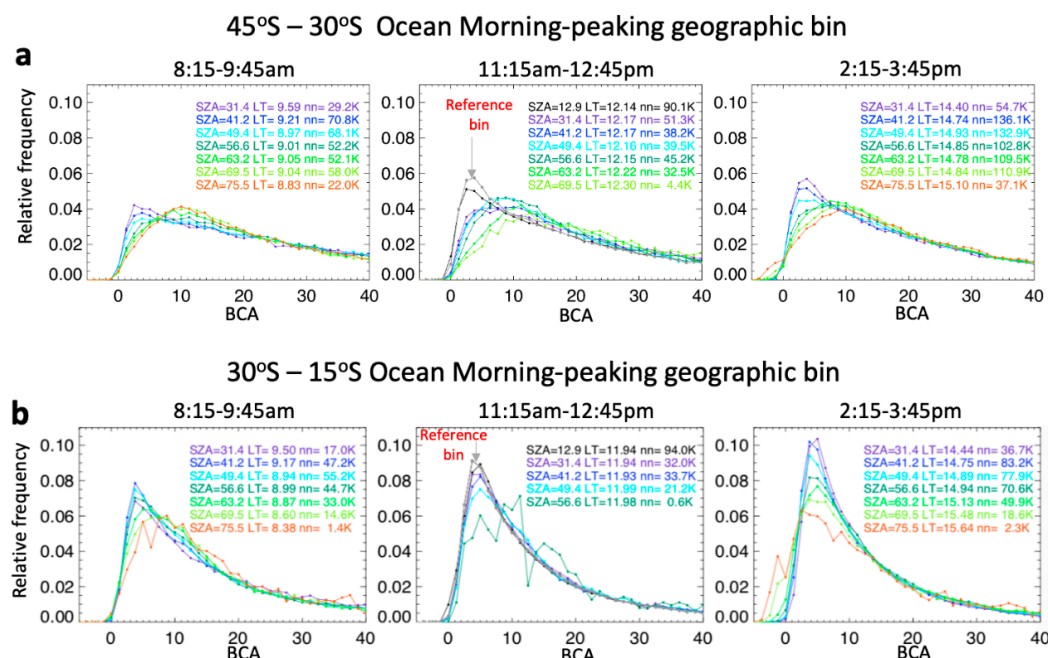

**Figure 3.** Relative frequency (histograms) of BCA for the ocean morning-peaking geographic bin in two latitude bands (**a**) 45°S to 30°S and (**b**) 30°S to 15°S. These areas are shown in Figure 2a by the red-colored cells over ocean. The left two panels are for observations sampled in the morning (local time), the middle panels close to noon, and the right panels in the afternoon. Within each local time bin, the observations are further categorized by solar zenith angle. The colored legends show the mean solar zenith angle, the average local time, and the number of samples (nn) for each bin. An example of a reference histogram, as shown by the gray trace in the local noon time bin, is used when the solar zenith angle at local noon for the day and latitude of the observation falls in the first bin (0.0° to 25.8° bin with mean of 12.9°); observations with a local-noon solar zenith angle that fall in other bins will be adjusted using a different reference distribution (see text). These two geographic bins have a large number of samples and yield smooth histograms even when segregated by solar zenith angle. Some geographic bins have so few observational samples that no diurnal corrections can be performed (e.g., afternoon peaking marine clouds in the 45°S to 30°S latitude band).

A frequency distribution of cloud optical depths inferred from nadir observations from the global area coverage (GAC) advanced very high-resolution radiometer (AVHRR) channel 1 (0.63 um) over marine stratocumulus clouds showed a similar shift to higher cloud optical depths with an increasing

solar zenith angle (Figure 8 of Loeb and Coakley [22]). While it is beyond the scope of this study to determine if unaccounted 3D radiative transfer effects are the cause of the solar zenith angle dependence, we offer it as a possible explanation. Our SBUV daily cycle of mean BCA for the "ocean morning-peaking category" (marine stratocumulus) geographic bin showed the highest BCA values in early morning (Figure 4a), which was consistent with previous observational studies. In a review paper of stratocumulus, Wood [26] states that a diurnal maximum in cloud thickness and liquid water path typically occur in the early morning hours. Using a worldwide climatology of surface observations from weather stations and ships, Eastman and Warren [27] report that land cumulonimbus clouds have minimum amounts in the morning and have a maximum at 4:30 p.m. (their Figure 4). While the diurnal cycle from our "land afternoon-peaking 30°N to 45°N" geographic bin agreed with the surface observations, our tropical land geographic bin did not, and needs further investigation (Figure 4b).

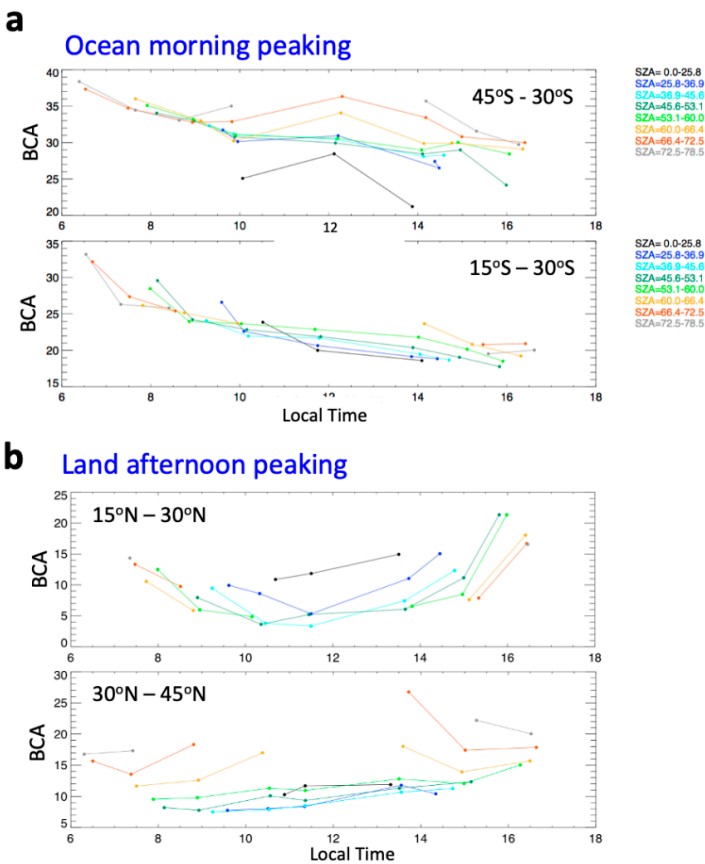

**Figure 4.** (**a**) The BCA daily cycle for clouds in the ocean morning-peaking geographic bin sorted by solar zenith angle. SBUV observations from red-colored grid cells over the ocean in Figure 2a are used. (**b**) The BCA cycle for land afternoon-peaking clouds. The SBUV observations from blue-colored grid cells over land in Figure 2a are shown here.

Our diurnal adjustment assumes that the percentile value ($P_{obs}$) of an observation $BCA_{obs}$ will not depend on the local time and solar zenith angle of the observation. $P_{obs}$ is the rank of an observation with respect to the others within its solar zenith angle and local time bin. The minimum BCA value in the bin will be associated with the zeroth percentile and the maximum BCA with the 100th percentile. The $P_{obs}$ are determined by summing the frequency distribution to obtain a cumulative frequency distribution for all the solar zenith angle and local time bins in a given geographic bin. Consistent with Equation (2), the diurnal adjustment, $\Delta BCA_{diurnal}$, is simply $BCA_{obs}$ minus the BCA of the reference distribution; $BCA_{ref}$ having a percentile value equal to the $P_{obs}$.

Figure 5 shows the $\Delta BCA$ diurnal adjustment for the same geographic bins shown in Figure 3. Basing the diurnal adjustment on percentile rank means that observations with low percentile rank

(low values of $BCA_{obs}$) will have negligible adjustments; instead, most of the adjustments will be for $BCA_{obs}$ with higher percentile rank. The "ocean, morning-peaking" 30°S to 15°S band, which consisted of low-level solid stratocumulus cloud decks, had positive values of $\Delta BCA_{diurnal}$ which were subtracted from observations taken in the morning; likewise, it had mostly negative values of $\Delta BCA_{diurnal}$ that were subtracted from evening-time observations (Figure 5b). Clouds in the 45°S to 30°S band were higher altitude and showed a strong solar zenith angle dependence, perhaps because they were more broken than the solid stratocumulus decks; they almost always had their BCA reduced regardless of the local time of observation (Figure 5a).

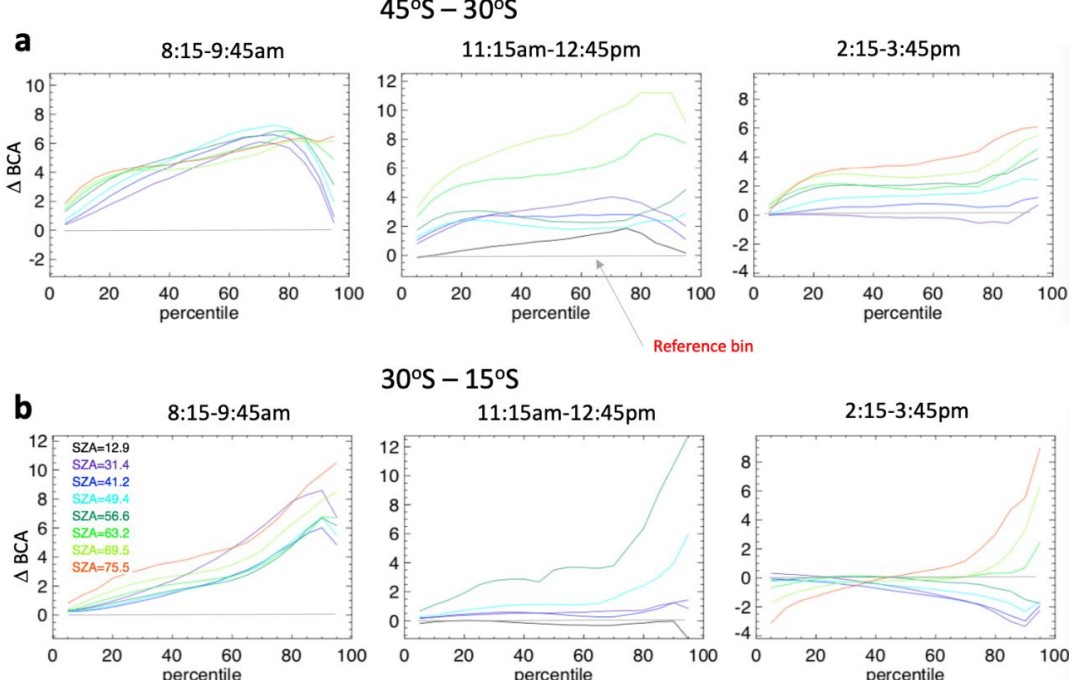

**Figure 5.** The diurnal adjustment ($\Delta BCA_{diurnal}$) for clouds in the ocean morning-peaking geographic bin for two latitude bands (**a**) 45°S to 30°S and (**b**) 30°S to 15°S, as function of the percentile of the observed BCA.

The inter-satellite differences were significantly reduced after the BCA were diurnally adjusted (Figure 1b). The high BCA observed in the morning were subsequently within two BCA of the afternoon measurements. A map of morning minus afternoon diurnally adjusted BCA shows small differences, indicating that the approach was effective (Figure 2b).

*3.3. Merging SBUV Instruments, OMPS*

Even after diurnal adjustment, the BCA time series of SBUV and OMPS mapper instruments over both ocean and land still showed small inter-satellite differences (Figure 6a,b), which still had to be considered when constructing a merged product. Our merging approach was to favor BCA values observed closer to a reference local time of noon, and to only use BCA values observed in early morning or late afternoon if no others existed. Figure 7 shows a time series of (a) absolute values and (b) monthly anomalies of our SBUV and OMPS merged product for seven latitude bands over ocean and land. Positive BCA excursions from the two volcanoes (black semi-circles on the horizontal axis) and El Niño events (green triangles) were readily seen in the tropical band 15°S to the equator.

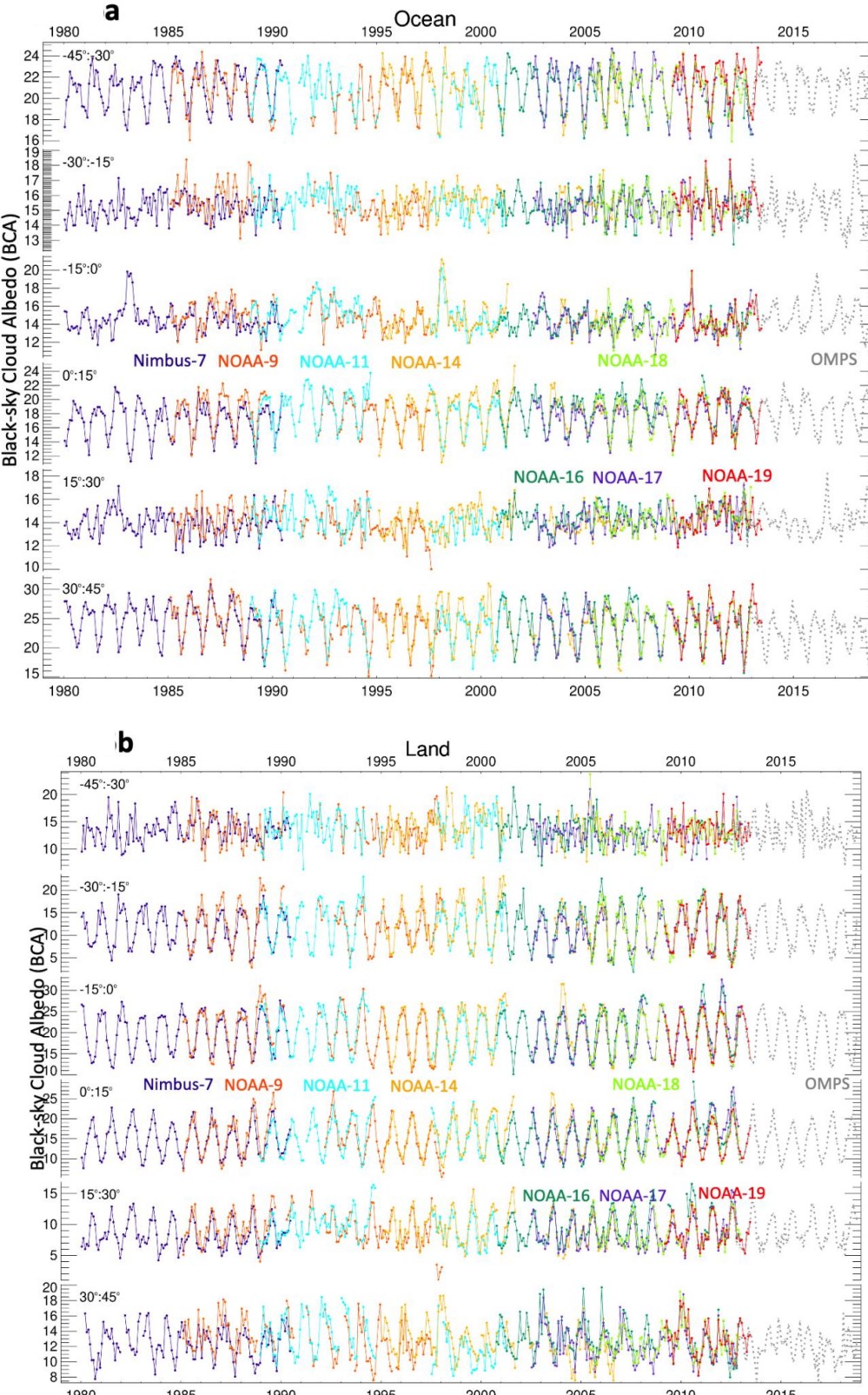

**Figure 6.** Monthly mean values of BCA (%) from the SBUV and the OMPS instrument for six 15°-wide latitude bands and from 45°S to 45°N over (**a**) ocean and (**b**) land.

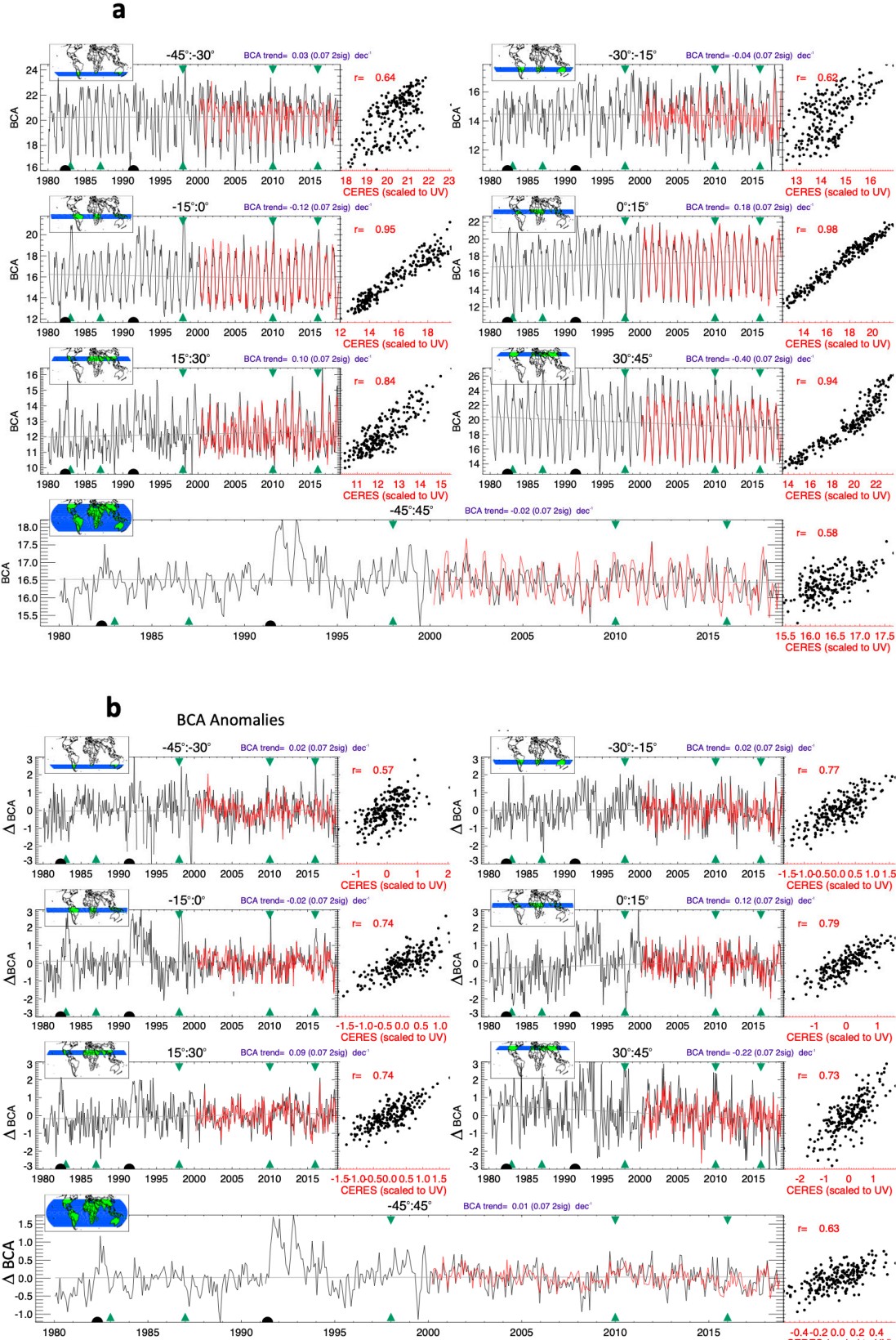

**Figure 7.** UV black-sky cloud albedo (%) (black trace) with shortwave cloud albedo from CERES (red trace) over ocean and land: (**a**) monthly values and (**b**) de-seasonalized monthly anomalies. The CERES SW cloud albedos have been scaled to be a proxy for the UV BCA. A single scaling factor for each 15° wide latitudinal band is used. In addition, on RHS there is a scatter plot of UV BCA versus CERES. Years with El Niño conditions (green triangles) and major volcanic eruptions (black semi-circles) are marked.

## 4. Comparison with CERES SW Broadband

The CERES TOA shortwave (0.3–5 μm) clear-sky and all-sky albedos used in this study are the monthly means of the single-scanner footprint energy balanced and filled (EBAF) TOA fluxes from Terra, Aqua, and NPP satellites (CERES_EBAF-TOA_Ed4; Loeb et al., 2018 [29]). Since we were interested in the TOA SW radiation from clouds, we used the difference between the clear-sky TOA SW flux minus the all-sky flux. The clear-sky product only includes forcing from water vapor and surface albedo, while the all-sky product includes all forcings. The difference of these two products (clear sky minus all sky), the SW cloud albedo radiative forcing, is largely sensitive to cloud properties and amount, but also aerosols in some locations. The cloud albedo forcing was normalized by the TOA SW downward flux (i.e., solar insolation) to produce the SW cloud albedo.

Besides having a much wider spectral bandwidth compared with those of our UV instruments, there were other important differences between the UV BCA and the SW CERES products. The CERES instrument observes directional radiances, like the OMPS instrument used in this study, which vary with the satellite viewing angle. The CERES albedo algorithm accounts for the BRDFs from different scene types. Angular directional models (ADM) are used to convert the directional radiances to fluxes (irradiance), depending on the scene of which each footprint is classified. For a given FOV, the appropriate ADM is chosen from a suite of +600 models based on the geo-type, imager cloud property, and atmospheric structure [4,30,31].

To facilitate comparison with the CERES record, we scaled the shortwave cloud albedo to the UV BCA. A scaling factor for each 15° latitude band was determined by dividing the zonal mean (e.g., 45°S to 30°S) UV BCA by the shortwave zonal mean; values ranged from 0.57 to 0.69. Time series of the SBUV and OMPS merged product (black trace) compared with the scaled CERES record (red trace) are shown in Figure 7. Latitude bands with a strong seasonal cycle (e.g., in the tropics Figure 7a) displayed the best agreement (r > 0.9). When the seasonal cycle was removed for the monthly anomalies, the correlations were reduced (r > 0.7), except for 45°S to 30°S (Figure 7b).

Despite their coarse temporal and spatial resolution, the SBUV instruments can capture the gross climatological patterns of global SW cloud albedo. As shown in Figure 8, the BCA from SBUV for July 2013 samples most of the large-scale cloud patterns seen by the CERES. Other months in the record also showed adequate coverage, unless too many of the observations were at high solar zenith angles (>75), in which case the month was discarded. Therefore, egregious spikes seen in some of the time series (e.g., Figure 7b, 15°S to 0° in 1998) are not from poor sampling, but are physically real.

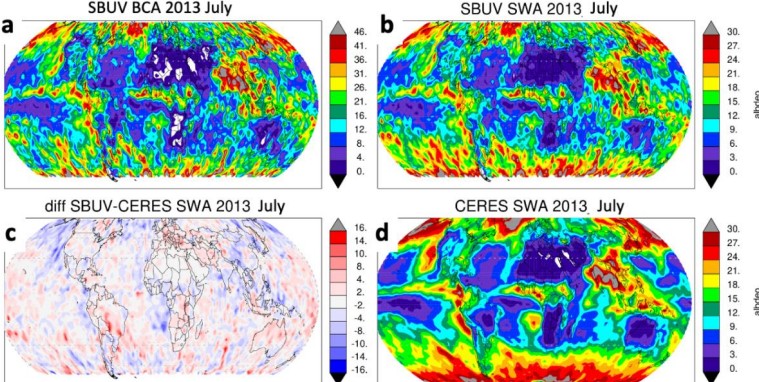

**Figure 8.** Map for July 2013 of (**a**) UV black-sky cloud albedo (%) from the SBUV merged product (NOAA-17 and -19 are the SBUV instruments operating at this time), (**b**) map of proxy-shortwave cloud albedo derived from UV BCA, (**d**) map of shortwave cloud albedo from CERES, (**c**) difference map of UV-derived proxy-SW cloud albedo (map (**b**)) minus CERES SW cloud albedo (map (**d**)).

For more precise comparison, we converted the UV BCA to a proxy shortwave cloud albedo using a set of empirically derived polynomials that varied with cloud phase (Figure 8b). These polynomials

were derived using the CERES cloud phase product which reported the water-to-ice ratio. There was a different polynomial fit between CERES SW cloud albedo and UV BCA for liquid water cloud conditions, mixed-phase, and almost pure ice conditions. We used a monthly climatology of cloud phase generated from the CERES record to determine which polynomial to use based on the month and the geographic location. The difference between our proxy SW cloud albedo and the CERES was fairly random and below 10% albedo for most locations (Figure 8c).

## 5. Cloud Responses

### 5.1. ENSO

Studies have found consistent cloud responses to ENSO variability among ship-based cloud data, CERES satellite data, and climate models. Using ship-based cloud observations, Park and Leovy [32] and Eastman et al. [33] both showed that interannual variations of cloud cover in the tropics have strong correlations to the ENSO index. The ENSO influences on the interannual variability of the cloud radiative effect were examined by Moore and Vonder Haar [34] and Kato [35] using CERES data.

By extending the global cloud observations back to 1980, we were able to present the UV cloud responses to five different ENSO events. We examined monthly BCA over two geographic boxes. The first region is over the Eastern Pacific (7°S to 7°N and 153°E to 100°W), where the warm SST anomalies were observed during a typical El Niño event. The second box is further west, where SST anomalies are neutral or below normal (20°S to 20°N and 95°E to 153°E), as shown in Figure 9a. A time series of monthly anomalies of BCA showed a strong positive anomaly over the warmer waters of the topical Eastern Pacific for each of the five El Niño events, as shown by green triangles. These anomalies occurred in the Januarys of 1983, 1987, 1998, 2010, and 2016, and consistently had a ΔBCA of +6 to +10 despite a +0.8 °C warming of global temperatures over the 39-year span. The 1992 anomaly was from the Mt. Pinatubo eruption. The multivariate ENSO index [36] (Wolter and Timlin, 1993) explains about half (r = 0.7) of the variability in the BCA anomalies over this limited geographic area. The result is consistent with the expectation of slowdown of the Walker circulation during El Niño and subsequent enhanced cloudiness (positive BCA anomalies) over the warmer ocean of the eastern Pacific. The BCA time-series from the second box showed a compensating reduction of cloudiness (negative BCA anomaly) during each El Niño event (right panel, Figure 9a) This is consistent with enhanced subsidence in the Western Pacific during El Niño conditions. Note the high correlation with the CERES SW cloud albedo (r > 0.90) over these tropical regions.

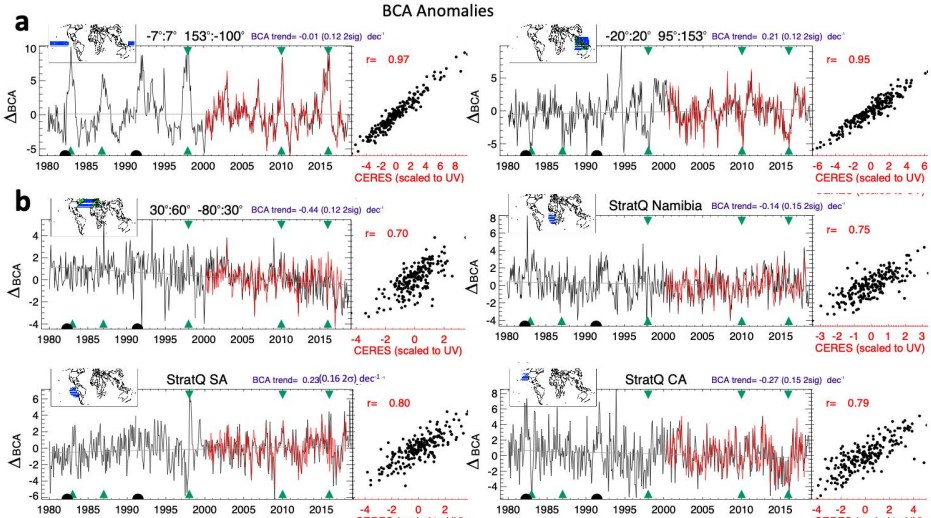

**Figure 9.** Monthly anomalies of UV BCA % (black trace) and shortwave cloud albedo from CERES (red trace) for selected geographic areas. (**a**) Regions with cloud response from ENSO; (**b**) the North Atlantic, and the three important stratocumulus decks off the coasts of Namibia, South America, and California.

### 5.2. Volcanoes

Apparently during each El Niño event, the positive cloud response from convection over the warm waters of the Eastern Pacific was compensated by a reduction in cloudiness elsewhere. The 45°S to 45°N BCA record showed no ENSO signal (see green triangles in lowest panel, Figure 7b). Instead, the most prominent features were coincident with the late March/early April 1982 El Chichón, and the early June 1991 Mt. Pinatubo volcanic eruptions (see black semi-circles, Figure 7b). The significantly larger and longer BCA perturbation for Mt. Pinatubo, compared with that of El Chichón, was consistent with the former's much larger $SO_2$ injection amount in the stratosphere (18 versus 8 megatons, Global Volcanism Program [37,38]). Both events lofted $SO_2$ high enough (>17 km) into the stratosphere where it could form sulfate aerosols, which remained there for 12–24 months. Currently, our BCA calculation does not account for aerosols, only C-1 water clouds, so they were treated as such. Although both volcanoes had similar latitudes of injection (15°N El Chichón and 17°N Pinatubo), maps of the eruption-induced BCA perturbation are significantly different (Figure 10). The El Chichón BCA perturbation was at least +1 throughout most of the NH tropics. This was consistent with aircraft lidar data that measured a latitudinal extent of 7°S to 37°N for the stratospheric aerosol cloud 6 months after the eruption [39]. In contrast, Mt. Pinatubo appeared to impact both hemispheres. Optical depths of aerosols observed by the stratospheric aerosol and gas experiment II (SAGE II) showed significant loading between 10°S to 30°N [40], which is consistent with our BCA perturbation map.

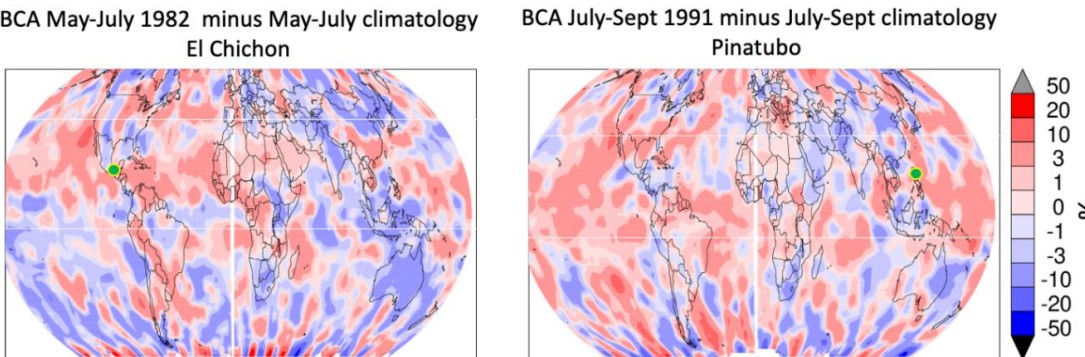

**Figure 10.** Perturbations of BCA induced by the El Chichón (**left**) and Mt. Pinatubo (**right**) volcanic eruptions. The global impact of the eruption emerges after the plume has been advected around the globe, which takes several weeks after the time of eruption. For El Chichón, which ended its eruption in early April of 1982, we average the months of May, June, and July of the that year and compare it with the May–July climatology over the entire record (1980–present). The same approach is used to determine the Mt. Pinatubo perturbation. For each volcano, the location of the eruption (green dots) and zonal extent of the $SO_2$ cloud (white lines) according to aircraft lidar measurements and stratospheric aerosol and gas experiment II (SAGE II) are shown.

### 5.3. Decadal Warming

While the BCA trend from 45°S to 45°N was negligible (lowest panel, Figure 7b), a map of BCA trends since 1980 shows there are geographic locations with significant changes in cloudiness (Figure 11). The largest and most widespread reduction in cloudiness occurs over the North Atlantic, while India and Southeast Asia show increasing cloudiness. Figure 9b shows trends for selected geographic areas with significant changes in cloudiness. Over the North Atlantic, there is a significant reduction in cloudiness throughout the record, and both CERES and the UV show an accelerated reduction after 2015. Three geographic areas dominated by marine stratocumulus are also shown. The trends show increasing cloud albedo off South America and reductions off the Californian and Namibian coasts. The features and trends of the scaled CERES SW cloud albedo anomalies (red) agree with our BCA record after 2000 for all geographic regions shown, and lend confidence to our BCA record.

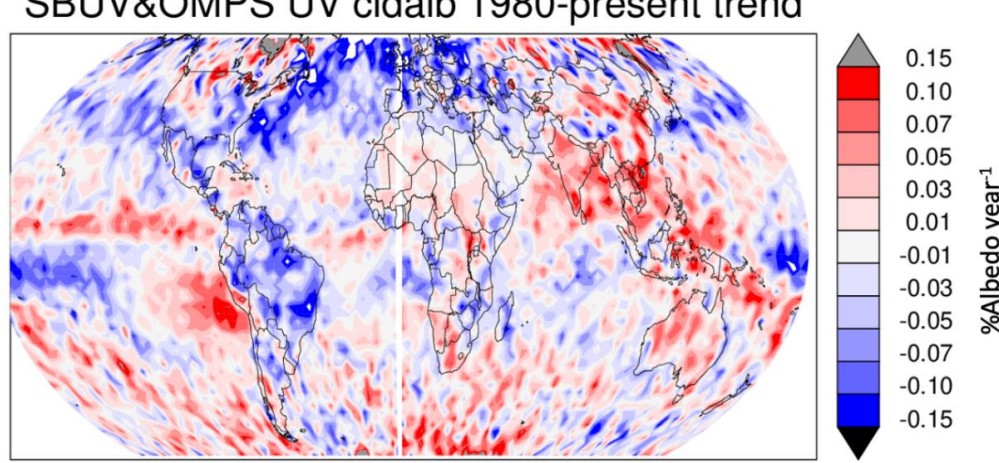

**Figure 11.** Map of UV BCA trends (1980–present) in % albedo year$^{-1}$. Linear trends are determined for each 2° latitude by 3° longitude grid cell.

## 6. Discussion

There are differences between the UV BCA and the CERES SW cloud albedo that merit discussion. The BCA used measurements at a single narrow-band wavelength (340 nm), while the CERES product was a broad-band, spectrally-integrated quantity. The BCA retrieval assumed that all clouds are a C1 water cloud, whereas the CERES product was able to more accurately account for differences in cloud phase. Despite these differences, our BCA record was well correlated with the CERES EBAF product. Measuring cloud albedo in the UV (away from ozone absorption) offered two advantages over SW measurements. Over land, the UV surface reflectivities are significantly lower than in the SW, so it is easier to isolate the cloud signal from the surface signal. Land UV reflectivities are generally between 2–4%, but can reach up to 14% in a few sandy desert regions [18]. Additionally, in the UV, it is easier to convert the observed directional radiances into flux irradiance. The significant Rayleigh scattering in the UV smears features in the anisotropy of an observed scene that usually appear more pronounced at longer wavelengths. Open ocean glint scenes are apparent in the UV, and we attempted to account for them in this study, but they are less distinct in the UV compared with longer wavelengths. Therefore, in the UV it is not as crucial to accurately characterize the scene anisotropy used to convert the observed radiance into fluxes. This may explain our agreement with the SW CERES cloud albedo despite the simple assumptions we made in our BCA derivation: A single droplet distribution, single cloud layer height, and a single phase (water). These advantages motivated Herman et al. [5], Weaver et al. [6], and now this work to derive a record of cloudiness from the suite of UV satellite instruments dating from 1980.

A companion paper by Weaver et al. [12] details our inter-satellite calibration method that uses intensities observed over the Antarctic and Greenland ice sheets. This technique estimated a 2σ uncertainty of 0.35% for the intensities, which translates to 0.6 for BCA. The number is an estimate of how well the instruments are inter-calibrated, but does not include errors made converting the intensities to BCA at a reference local time and solar zenith angle. The diurnal adjustment procedure, the plane parallel radiative transfer model, and its configuration with a C-1 cloud model, are all approximate representations of the real atmosphere and, therefore, introduce additional errors. We estimated a "satellite overlap" uncertainty from these errors by considering the spread of BCA values when multiple satellite instruments are observing the same month and 15° latitude band. The 2σ overlap uncertainty of about 1 BCA was larger than the calibration uncertainty of the UV instruments and was used to estimate the trend uncertainties discussed below. With this discussion of our CERES-validated product and the uncertainty analysis, we provide some additional comments on the BCA trends.

Our negligible BCA trend of 0.01 decade$^{-1}$ (45°S to 45°N) appeared to disagree with the quite significant −0.24 LER decade$^{-1}$ (60°S to 60°N) estimate of Herman et al. [5]. Their LER was produced using SBUV intensities calibrated only over Antarctica. The intensities used to derive BCA in our study were re-calibrated using observations over both the Antarctic and Greenland ice sheets [12]. We used the Greenland ice sheet observations to help detect so-called hysteresis error with the photo multiplier tubes (PMT). The PMTs on the earlier SBUV instruments were not able to quickly respond to the 4 orders of magnitude signal changes that occur when the satellite first comes out of darkness on each orbit and the instrument is again illuminated. This so-called hysteresis error is most severe for Nimbus-7 when it sees its first light over Antarctica; it then lessens along its orbit so that when it reaches Greenland the PMT has "warmed up" and responds consistently to illumination along the flight path. The intensities used in the Herman study were corrected for hysteresis [14]. Furthermore, in an attempt to remove observations potentially impacted by hysteresis, only observations with solar zenith angles below 70° were used in their Antarctic calibration. However, in our study we could not get the Nimbus-7 intensities to agree over both Antarctica and Greenland with those from NOAA-9 and -11 during the years when these instruments overlapped. We attributed this to residual hysteresis errors in Nimbus-7 intensities over Antarctica, after the initial correction was applied, and therefore limited our calibration region to Greenland. All other SBUV instruments were calibrated over both ice sheets. This revised approach resulted in minor changes to the calibration of the SBUV instruments used in the Herman study, with the important exception of Nimbus-7 SBUV; we reduced its intensities by 0.87% (Table 1 [12]). In an attempt to reproduce the Herman study, we calculated the BCA trend using their calibrated SBUV intensities that were done only over Antarctica. The resulting −0.14 decade$^{-1}$ (45°S to 45°N) trend was closer to their reported result. Despite the differences in calibration, our map of BCA trends agreed qualitatively with their LER trend map, as shown in their Figure 13. They showed the same reduction in cloudiness over the North Atlantic and increases over India and Southeast Asia.

Before developing our diurnal adjustment scheme, we did not know if the inter-satellite differences (Figure 2a) were caused by the cloud diurnal cycle or SBUV instrument issues (e.g., calibration or hysteresis). Our diurnal adjustment showed that the diurnal cloud cycle and the solar zenith angle of the observation explain much of these differences, but the adjustment scheme did not drive the trends. Trends determined using non-adjusted BCA values were qualitatively similar to those using diurnally adjusted BCA values. There were enough observations sampled near local noon time at the beginning, middle, and end of the 39-year UV record to capture an accurate trend.

Low altitude clouds, rather than their higher-level brethren, are the main contributors to the spread in net cloud feedback among different climate models [41]. Existing long-term observational records of low-level cloud amount (LCA) were sourced from the International Satellite Cloud Climatology Project (ISCCP) and Pathfinder Atmospheres-Extended (PATMOS-x). Seethala et al. [42] report trend maps (1984–2009) of LCA in their Figure 1; similar figures are shown in Norris et al. [43]. In agreement with our UV record, ISCCP and PATMOS-x both reported decreasing cloudiness over the North Atlantic, midlatitude North Pacific, and south of the equator in the Pacific. Geographic locations also in agreement that show increasing cloudiness are the tropical West Pacific, Northern Indian Ocean, and subtropical Southeast Pacific. However, there are disagreements; the ISCCP-Patmos-x record shows increasing LCA for three marine stratocumulus cloud regions off the coasts of South America, California, and Namibia. While our BCA record agrees with the trends over South America, we did not detect increasing cloudiness for the other two geographic areas. Instead, we showed reduced BCA off Californian and Namibian coasts. The different time spans of the records (ISCCP-PATMOS-x ends in 2009) may contribute to these disparities and needs to be further investigated.

## 7. Summary

The UV reflectance observed from low-Earth-orbit satellites provides a unique cloud albedo measurement in the middle and lower troposphere where clouds are highly reflective. This UV cloud

albedo correlates well with the CERES TOA cloud albedo on a monthly basis during the period in which they overlapped.

The newly calibrated and CERES-validated UV BCA data have produced, for the first time, a long-term UV cloud record since 1980, which enables a comprehensive comparison study of cloud and aerosol shortwave radiative responses to ENSO, volcanic forcing, and decadal warming. Five major ENSO events and two volcanic eruptions were covered within this long-term record of UV BCA measurements.

El Niño events increase cloudiness in the tropical Eastern Pacific and decrease cloudiness elsewhere in the tropics. These perturbations cancel out in the zonal mean, such that the El Niño signal was not apparent. Instead, the strongest signals in zonal means equatorward of 45° were the increase of albedo from the volcanic eruptions of El Chichón and Mt. Pinatubo.

Although our BCA record suggests no trend in cloudiness from (45°S to 45°N), we saw statistically significant reductions over the North Atlantic and the marine stratocumulus decks off the coast of California. Increases in cloudiness were observed for cloud decks off the coast of South America and over Southeast Asia.

**Author Contributions:** Conceptualization, C.J.W.; formal analysis, C.J.W.; methodology, D.L.W., P.K.B., G.J.L. and D.P.H.; resources, D.L.W. and P.K.B.; writing—original draft, C.J.W.; writing—review and editing, D.L.W., G.J.L., and D.P.H. All authors have read and agreed to the published version of the manuscript.

**Funding:** This work was supported by a NASA Making Earth System Data Records for Use in Research Environments (MEaSUREs) Project and the NASA Long-term measurement of Ozone program.

**Acknowledgments:** We appreciate the comments of the two reviewers and productive discussions with Jay Herman and Richard Stolarski.

**Conflicts of Interest:** The authors declare no conflict of interest.

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
