# Peer review of "A Long-Term Cloud Albedo Data Record Since 1980 from UV Satellite Sensors"

_remotesensing, doi:10.3390/rs12121982_

Round 1
Reviewer 1 Report
Review of “A long-term cloud albedo data record since 1980 from 3 UV satellite sensors” by C. Weaver, D. L. Wu, P. K. Bhartia, G. Labow and D. P. Haffner (remotesensing-821565)
Recommendation: minor revision
General comments: This is an excellent study that describes the methodology for calculating black-sky cloud albedo (BCA) from satellite measurements of UV reflection and combining measurements from multiple satellites over a four-decade period into a single relatively homogeneous dataset. Various interesting results are presented that demonstrate that the BCA climatology and variability is associated with real physical climate phenomena. It is clever to homogenize the satellite record by matching percentiles within latitude zones and adjusting for local time and solar zenith angle, thought this will incorporate some diurnal and seasonal effects along with retrieval effects. I don’t have much to criticize in the manuscript except to suggest some possible improvements.
Specific comments:
1) Line 67: why not a diurnal average?
2) Line 108: cloud amount or cloud optical depth?
3) Line 124: isn’t two samples per month quite small? Are there any temporal variations in what grid boxes of the zonal band are sampled?
4) Line 130: 5% BCA?
5) Figure 2b: It looks like the diurnal correction does not work so well over NH midlatitude land? Any ideas why this is?
6) Figure 3 and related: looking at variations in SZA for the same interval of local time in the same latitude zone will be sampling different parts of the seasonal cycle. There is a real seasonal cycle in cloudiness, as demonstrated by other cloud datasets, so this approach will mix albedo dependence on solar zenith angle with seasonal variations in cloud. Solar zenith angle will also vary with latitude in a latitude zone. Have the authors quantified how much solar zenith angle varies simply by the change in latitude across the zone, or across the interval in local time, or across the seasonal cycle and how this compares in magnitude to the various solar zenith angle intervals categories considered?
7) If adjustments are done by matching percentiles, does that mean that unphysical values are guaranteed to never occur, like might occur if subtraction of an offset causes a small positive value of albedo to become negative?
8) When merging the time series from various satellites, won’t choosing only the value closest to noon when available and late morning/afternoon values only when no others are available result in discontinuities? For example, perhaps two satellites have some bias offset with respect to each other. If one is closer to noon and the other closer to late afternoon, once one satellite drops out there will be a sudden artificial shift. An alternative approach would be to choose one satellite as a reference time series and then apply an offset to each other satellite time series that overlaps with the reference time series such that there is no average difference between them during the period of overlap. This can then be extended to all satellites over the whole time period.
9) The agreement between the BCA time series and CERES is encouraging, as is also the realistic relationship to phenomena like El Nino.
10) Lines 476-481: I find it encouraging that there appears to be substantial agreement between the spatial pattern of BCA trends in Fig. 11 and the spatial pattern of cloud amount trends in Fig. 1ab of Norris et al. (2016) – for example, decreasing cloud over the North Atlantic, midlatitude North Pacific, and south of the equator in the Pacific, and increasing cloud over the tropical west Pacific, northern Indian Ocean, and southeast subtropical Pacific. This suggests to me that we can observationally document cloud changes occurring over decades using independent datasets. Note that there was an extreme reduction of low cloud over the northeast subtropical Pacific years after 2009, and this may account for the difference in trend sign between this study and Seethala et al. (2015).
https://agupubs.onlinelibrary.wiley.com/doi/full/10.1029/2018GL078242
https://agupubs.onlinelibrary.wiley.com/doi/abs/10.1029/2019GL086705
Reviewer 2 Report
Overall the paper presents a thorough analysis of long term data from several platforms to explore large spatial scale trends in cloudiness. The results are supported with some statistical estimates of error in the calculations. One area that is discussed is the discrepancy between the Herman (2013) work and this study (e.g. line 443 et seq). This difference in mid-latitude trends is sufficiently important to climate modeling that it should be resolved with regard to the calibration methods used beyond the authors discussion. Is there a way to check the Herman results by looking again at the NIMBUS 7 data in the authors' analysis using Antarctic surface albedo separately from the Greenland-Antarctica mix? It would seem useful to see if the Herman results can be reproduced with their Antarctic albedo method?
I found the time series figures (e.g. Fig. 7 and 9 difficult to read even with enlarging the figures. It would help if the authors could use bold arrows rather than tics for the ENSO periods and perhaps larger blips for the volcano eruptions. Perhaps a shaded section could be used to identify change with ENSO and eruptions? In Figure 10, the location of El Chichon and Pinatubo could be better marked visually. In these figures the cloudiness contrasts seem distinct for the volcano locations even though the atmosphere was seemingly well mixed. Any author comment?
